# The Economic Burden of Prostate Cancer in Antigua and Barbuda: A Prevalence-Based Cost-of-Illness Analysis from the Healthcare Provider Perspective

**DOI:** 10.3390/ijerph21111527

**Published:** 2024-11-18

**Authors:** Andre A. N. Bovell, Cebisile Ngcamphalala, Adrian Rhudd, Jabulani Ncayiyana, Themba G. Ginindza

**Affiliations:** 1Discipline of Public Health Medicine, School of Nursing and Public Health, University of KwaZulu-Natal, Durban 4000, South Africa; cn2323@cumc.columbia.edu (C.N.); ncayiyanaj@ukzn.ac.za (J.N.); ginindza@ukzn.ac.za (T.G.G.); 2Urology Department, Sir Lester Bird Medical Centre, Saint John’s 4586, Antigua and Barbuda; adrian.rhudd@msjmc.org; 3Cancer & Infectious Diseases Epidemiology Research Unit (CIDERU), College of Health Sciences, University of KwaZulu-Natal, Durban 4000, South Africa

**Keywords:** Antigua and Barbuda, prostate cancer, cost of illness, economic burden, direct medical cost

## Abstract

In Antigua and Barbuda, prostate cancer is known for its epidemiological burden; however, its economic burden on the healthcare system is unknown. This study aimed to assess the economic burden of prostate cancer in Antigua and Barbuda from the healthcare provider’s perspective. To conduct this prevalence-based cost-of-illness study, we used patient data abstracted from records at key study sites for the period of 2017–2021 to establish a yearly prevalence. Top-down and bottom-up approaches were used to estimate the direct medical cost. The cost was computed at the 2021 price level and converted to United States dollars (USD). The total annual direct medical cost for prostate cancer was estimated at USD 1.8 million (ranging between USD 1.4 million and USD 2.3 million). Stages II and III disease accounted for a combined greater share of the cost. The direct medical unit cost for screening, diagnosing, and treating a prostate cancer patient was USD 126,388.98. The top contributors to this cost were surgery (USD 20,913.42), renal complications/renal failure (USD 20,674.86), and hormonal therapy (USD 31,824.00). The results of this study provide evidence of the economic burden of prostate cancer in Antigua and Barbuda. Our findings appear reasonable. Besides contributing to further economic research, they will be useful for policy development, resource allocation, and cost containment measures.

## 1. Introduction

Prostate cancer is a serious health concern in many countries [1]. It is the second most commonly diagnosed cancer in males and the fifth most prevalent cause of cancer-related mortality among men, accounting for 14.1% (n = 1,424,100) of new cancer cases and 6.8% (n = 374,000) of cancer-related deaths among men worldwide in 2020 [1]. In Antigua and Barbuda, prostate cancer is the leading type of cancer in terms of incidence and mortality in men, with age-standardized rates of 69 per 100,000 and 53 per 100,000, respectively, for the period of 2001–2005 [2,3].

In addition to the well-documented epidemiological burden of prostate cancer, there is increasing evidence of its economic impact worldwide, particularly on health systems in many countries. For example, Beaulieu et al. reported that in 2009, prostate cancer accounted for approximately USD 24 billion out of the estimated global cost of USD 154 billion for treating cancers [4]. In the United Kingdom, the total annual cost for the first year of prostate cancer treatment was estimated at EUR 117 million [5]. In Iran, Mojahedian et al. reported this cost to be USD 2900 million [6], while in Eswatini, it was USD 6.2 million [7]. Notably, evidence of the economic burden of prostate cancer in countries in the English-speaking Caribbean, such as Antigua and Barbuda, is limited.

Given the projected increase in annual prostate cancer cases in Antigua and Barbuda in the years to come [1], there will be an expected and corresponding need for country-specific evidence of its economic burden, the details of which can provide insight into how costs are distributed across the resources required to treat and manage this type of cancer [8].

The purpose of this study, therefore, was to estimate the economic burden of prostate cancer in Antigua and Barbuda from the healthcare provider’s perspective.

## 2. Materials and Methods

### 2.1. Study Area and Population

Antigua and Barbuda is the largest of the English-speaking countries of the Leeward Islands [9]. It is a twin-island state, with Antigua, the larger of the two, located 650 km southeast of Puerto Rico, and Barbuda, the smaller island, located 48 km north of Antigua [10]. The country’s population at its last census, held in 2011, was estimated at 85,567, with 40,986 males and 44,581 females and 80% (n = 68,702) of the population being between the ages of 15 and 74 years of age [11].

As a member of the Eastern Caribbean Currency Union (ECCU), Antigua and Barbuda uses a common currency pegged to the United States dollar, i.e., the Eastern Caribbean dollar (XCD) [12,13].

With its monetary policy generally guided by the Eastern Caribbean Central Bank (ECCB), Basseterre, St. Kitts and Nevis, fiscally, one of the main contributors to the country’s gross domestic product (GDP) is tourism [14]. The country’s GDP per capita (current USD) was estimated at USD 17,178.50 for 2021, and the health expenditure per capita in the same year was approximately USD 923.41 [15].

Public healthcare is chiefly financed by the government-funded Medical Benefits Scheme (MBS), Antigua and Barbuda and allocations from the Ministry of Finance, Antigua and Barbuda. Cancers are primarily diagnosed and managed at the country’s lone tertiary care hospital [16,17].

The study population consisted of men aged 18 years and older who had been diagnosed with prostate cancer between 1 January 2017, and 31 December 2021. The cancer cases were classified according to the International Classification of Diseases, 10th version (ICD-10) codes (C61) [18].

### 2.2. Inclusion and Exclusion Criteria

In the absence of a cancer registry, information about prostate cancer cases diagnosed between 1 January 2017 and 31 December 2021 was obtained from medical records at Sir Lester Bird Medical Center, Antigua and Barbuda’s lone tertiary hospital; the Cancer Center of the Eastern Caribbean (TCCEC), a public/private cancer facility; and the Medical Benefits Scheme (MBS), a statutory health organization. Information about prostate-cancer-related deaths was obtained from the Health Information Division, Ministry of Health, Antigua and Barbuda.

Cases of recurrent disease were excluded from the study.

### 2.3. Management of Prostate Cancer in Antigua and Barbuda

Oncological care in Antigua and Barbuda is broadly informed by cancer control and prevention strategies suggested by the World Health Organization (WHO) [19]. On this premise, adaptations of the National Comprehensive Cancer Network (NCCN) clinical practice guidelines in oncology are used in the treatment of many locally occurring cancers [20]. These guidelines provide standard information about optimizing care and improving management outcomes with respect to the screening, diagnostic workup, staging, and treatment of cancer patients [21]. Cancer management may require varying degrees of inpatient and/or outpatient hospital care, based on the cancer type and individual patient needs.

Prostate cancer management may start with a male patient consulting a general practitioner. Based on the clinical findings, obtained via asymptomatic or symptomatic screening, the patient may be referred to a urologist (Figure 1). At times, a patient may choose to consult a urologist directly, particularly if the patient is ≥40 years of age or has a family history of prostate cancer. The urologist conducts an array of screening activities: (i) clinical assessment, (ii) a digital rectal exam (DRE), and (iii) a blood test (prostate-specific antigen (PSA) test) [22,23]. If the DRE is abnormal (palpation of irregularity or nodules) [24,25] and the PSA level elevated, a needle or transrectal ultrasound biopsy (TRUS) is performed [22].

Should the results of the biopsy return as positive, the patient’s disease is then staged. This involves the use of plain X-rays and computer tomography scans, with contrast scans of the abdomen and pelvis [22,23]. Collectively, this determines whether the disease is localized or metastatic. The treatment for localized prostate cancer depends on (i) the stage and extent of the disease, (ii) the age of the patient, and (iii) the patient’s general health. Thus, it may involve (a) active surveillance, (b) watchful waiting, (c) surgery (radical prostatectomy), (d) radiotherapy (external beam radiotherapy), or (e) androgen deprivation hormonal therapy [22,23]. Patients diagnosed with metastatic disease are referred to a medical oncologist for care, which may involve radiation therapy, hormonal therapy, or systemic therapy. Robotic prostatectomy, brachytherapy, bone scanning, positron emission tomography (PET) scanning, and prostate multiparametric magnetic resonance imaging (MRI) are not currently available in Antigua and Barbuda [22]. Cancer patients requiring specialized intervention, such as robotic prostatectomy, may be referred to other countries, and the care may not be supported by the Medical Benefits Scheme.

In our study, the cost estimates for all care components involved in the management of prostate cancer were based on the 2021 market prices, as per the sources (expressed in terms of private clinics and laboratories) and the SLBMC unsubsidized prices (Table 1).

### 2.4. Cost

This was a prevalence-based cost-of-illness (COI) study conducted from the healthcare provider’s perspective [26]. Healthcare interventions required in the prostate cancer management continuum (screening, diagnosis, and management) were identified (micro-costing approach), quantified, valued per case, and extrapolated using prevalence data to estimate national costs. Data were collected retrospectively using patient charts and an Excel form.
Direct Medical Costs of Disease (mc)=∑(Mi×Pi) 

*Mi* is number of cases requiring healthcare;*Pi* is the unit costs of required healthcare resources per case;*mc* is the total costs.

The direct medical costs considered recurrent costs [7], which included the amount spent on personnel, travel, consumables, supplies (medical and non-medical), treatment, administration, and overheads [7]. For our cost analysis, we used an estimate of the number of prostate cancer cases in a single year in Antigua and Barbuda, calculated by subtracting the number of deaths from the total number of cases and dividing the answer by 5 [27].

### 2.5. Cost Data

Direct medical costs related to the screening, diagnosis, treatment, and follow-up care of prostate cancer were collected using an electronic datafile (Excel spreadsheet) specifically designed to compile cost data according to on-site datafiles and routine records available at the study sites [28]. This data collection tool was developed based on the prostate cancer screening, diagnosis, and general treatment pathway used in Antigua and Barbuda (Figure 1).

The aforementioned direct medical costs were obtained from reimbursement receipts submitted to the Medical Benefits Scheme as part of the financial claims applications for laboratory, pathology, imaging, surgical, ultrasound, hospitalization, medication, and other similar services provided to beneficiaries by private health facilities on the island [17]. These records are archived in the Revenue and Transactions Services departments of the Medical Benefits Scheme [17]. Additionally, cost information was collected from private clinics and laboratories.

The average cost of each intervention, namely active surveillance, watchful waiting, surgery, radiation therapy, hormonal therapy, and systemic therapy, was multiplied by the corresponding number of patients who received that intervention. Using the 8th Edition of the *American Joint Committee on Cancer Staging Manual* (*AJCC 8th edition*) guidelines for staging the disease, we assumed that all men with prostate cancer were screened and diagnosed according to the pathways described in Figure 1 [29].

Costs were reported in USD, obtained following adjustment using the country consumer price index (CPI) of 2021 and the USD 2021 exchange rate (1 USD = 2.7169 XCD) [7,30], as follows:$$Value\;in\;2021\;USD=Base\;year\;price\times \frac{CPI\;in\;2021}{CPI\;in\;based\;year}$$

CPI in 2021 = 95.27; CPI in based year = 95.27 [31].

Total costs were computed as follows:Direct medical costs of disease (mc)=∑(Mi×Pi) 

Given that a PET scan, brachytherapy, and robotic prostatectomy, including their corresponding transportation and accommodation costs, constitute significant healthcare imports that comprise services financed by the Medical Benefits Scheme for deserving patients, the total costs were presented with and without the inclusion of these care components [32].

### 2.6. Sensitivity Analysis

A sensitivity analysis was performed to investigate the impact of the uncertainty of key parameters included in our cost-of-illness model [26]. Costs were varied using the lower and upper bounds of ±25% [7].

### 2.7. Ethical Considerations

Approval for this study was granted by the Antigua and Barbuda Institutional Review Board, Ministry of Health (AL-04/052022-ANUIRB); the Institutional Review Board of Sir Lester Bird Medical Center; and the University of KwaZulu-Natal Biomedical Research Ethics Committee (BREC/00004531/2022). Data on our population of prostate cancer cases for the study period of 2017–2021 were accessed only by the principal investigator, who is also the corresponding author of this paper, from 16 September 2022, to 16 January 2023, at the study sites. Additionally, all cost data, based on the 2021 prices, were collected by said author from 22 November 2022, to 25 January 2024. This study did not involve direct contact with cases, and there was no direct risk to anyone [33]. We ensured de-identification and anonymization by not recording the names of any of the patients at, during, or after data collection [33].

## 3. Results

### 3.1. Background Population of Cases

Table 2 shows that the mean age at which the diagnosed individuals were seen by a specialist was 66.5 years. Almost 82% of men were within the 55-to-74-year age bracket, while a mere 3% were 45 to 54 years old (Table 2). Approximately 62% of cases resided in the most populous parish of St. Johns, with zero cases recorded for the parish of Barbuda.

### 3.2. Estimate of Prostate Cancer Cases in a Single Year

The information obtained indicated that 13 of the 109 diagnosed cancer cases had died during the study period. We therefore estimated that there were 19 cases of prostate cancer, on average, in a single year in Antigua and Barbuda. This was calculated by subtracting the 13 deaths from the 109 cases diagnosed in the period and dividing the result (96) by 5.

### 3.3. Direct Medical Unit Costs

The results showed that the overall estimated direct medical unit cost for screening, diagnosing, staging, treating, and managing a patient locally in 2021 was approximately USD 126,388.98 (adjusted), with this cost increasing slightly to USD 133,475.56, if robotic prostatectomy replaced the locally available open prostatectomy as the surgical option (Table 3). The leading contributor to the overall direct medical unit costs was treatment (USD 70,506.58 with open prostatectomy as the surgical option and USD 77,593.16 with robotic prostatectomy as the surgical option). The other top contributors to the overall direct medical unit costs were post-treatment side-effects relieving care (USD 33,346.45) and other direct costs (USD 16,968.51), in that order (Table 3). The direct medical unit costs for prostate cancer by clinical stages ranged from lows of USD 83,901.85 and USD 89,252.77 for stages I and IV, respectively, to highs of USD 121,626.19 and USD 123,166.19 for stages II and III, respectively (Figure 2A, Table 3). These estimates increased marginally for stages II and III of the disease, if robotic prostatectomy replaced the locally available open prostatectomy as the available surgical option (Table 3). The main drivers of the overall direct medical unit cost for treatment were hormonal therapy (USD 31,824.00), surgery (USD 20,913.42 for open prostatectomy and USD 28,000.00 for robotic prostatectomy), and radiotherapy (USD 12,974.35) (Figure 2B). Other treatment components (including biopsy prophylaxis) combined accounted for USD 1964.65, while systemic therapy also contributed to the direct medical unit cost of treatment, at USD 3019.26 (Figure 2B). Most of the direct medical unit costs related to post-treatment side-effects relieving procedures were the costs of managing renal complications/renal failure (USD 20,674.86) and other complications of treatment (USD 7448.14) (Figure 2C, Table 3). All other components of the post-treatment side-effects relieving procedures collectively amounted to total costs of USD 5223.46 (Figure 2, Table 3). Regarding other direct costs, the healthcare import of brachytherapy was a dominant cost driver at USD 13,000.00 (Figure 2D, Table 3).

### 3.4. Total Annual Estimated Direct Medical Costs

The total annual direct medical cost for prostate cancer was estimated to be USD 1,820,510.70 (ranging between USD 1.365,383.03 and USD 2,275,638.38), which included healthcare imports. The treatment of clinical stages I to IV accounted for the largest share of the annual direct medical costs, at 86% (USD 1,566,642.66; range USD 1,174,982.00 to USD 1,958,303.33) (Table 4). The second major contributor to the annual direct medical costs was post-treatment side effects relieving procedures, accounting for 5% of the costs (USD 94,605.32; range USD 70,953.99 to USD 118,256.65) (Table 4). The next highest contributor to the annual direct medical costs was diagnosis, accounting for approximately 4% of the costs (USD 67,738.04; range USD 50,803.53 to USD 84,672.55). The remaining care components combined accounted for almost 5% of the annual direct medical costs (USD 91,524.68; range USD 68,643.51 to USD 114,405.85) (Table 4, Figure 3). Regarding the “other direct cost” parameters, overseas brachytherapy, with a suggested low patient volume, was the highest cost driver, followed by local transportation services. For follow-up care, post-treatment consultations were the leading driver of the annual direct medical costs, accounting for 60% of the annual direct costs linked to this component of care.

When healthcare imports were excluded from the analysis, the total annual direct medical cost was estimated to be USD 1,785,836.10 (range: USD 1,339,377.08 to USD 2,232,295.13). The effect of this was an obvious decline of USD 34,674.60 in the estimates for the “other direct costs” parameter.

## 4. Discussion

This study reported the costs of prostate cancer in Antigua and Barbuda by estimating the direct medical costs from the healthcare provider’s perspective. The estimated total annual direct medical cost or economic burden of prostate cancer in 2021 was USD 1.8 million (ranging between USD 1.4 million and USD 2.3 million), with a nominal reduction of USD 0.03 million when the healthcare imports of PET scans and brachytherapy and their associated transportation and accommodation costs were excluded from the analysis. The main driver of the total annual direct medical costs was treatment, which accounted for 86% (USD 1,566,642.66) of the total annual costs, an observation consistent with that of studies conducted elsewhere showing that treatment can account for upwards of 70% or more of total costs, as reported, for example, in Eswatini, a low- to middle-income country [7]. In the absence of similar cost studies performed in other Eastern Caribbean Currency Union member states with which comparisons could have been made, this finding, while reasonable, appears to suggest that the annual direct medical costs of prostate cancer represent a significant financial burden for Antigua and Barbuda’s healthcare system, particularly when viewed in the context of its current health expenditure per capita, which was USD 923.41 in 2021 [15]. In general, the total annual direct medical costs were approximated to 2% of the budgeted national health allocations for 2021 [34].

Our diagnostic costs appeared reasonable and accounted for almost 4% (USD 67,738.04) of the annual direct medical costs. Coupled with screening costs, at <1% (USD 5839.46) of the annual direct medical costs, this underscores the need to either maintain or reduce these low-cost parameters in the cost containment measures in the foreseeable future. Regarding components of care, the direct medical costs of treating prostate cancer by cancer stage varied, with stages II and III recording higher costs compared to stages I and IV. Although radiation and hormonal therapy were among the main cost drivers in all stages of prostate cancer, in stages II and III of the disease, the main cost drivers also included surgery and systemic therapy, with hormone therapy, surgery, and radiotherapy being the costliest treatments. This observation is consistent with the observations of studies conducted in high-income countries that suggest that radiotherapy, surgery, and hormonal therapy account for the greatest per capita costs of treating prostate cancer [35]. Additionally, the observed differences in treatment costs by stage are consonant with the general cost increases anticipated with an advanced stage at diagnosis and disease progression [36]. The post-treatment side-effects relieving care was another key contributor to costs, accounting for about 5% of the total annual direct medical costs, with one of the leading cost drivers being renal complications/renal failure, a common condition known to be prevalent in most elderly men with advanced disease [37,38]. In most of the cases in our study, prostate cancer occurred between ages 55 and 74 years.

Of note was the absence of orchiectomy as a treatment option in this country, as well as in our study estimates. Consistent with previous studies, this could possibly be explained by (i) patients having a greater say in the treatment choices available to them and/or (ii) hormonal therapy being preferred as a less problematic treatment option in surgery-shy and/or high-risk patients [39].

Observably, several studies have estimated the economic burden of prostate cancer over time [6,7,18]. The findings of these studies vary, depending on the methodology, underlying population characteristics, geographic region, and standards of practice guidelines for prostate cancer management [5,7,36]. Even though our study estimates would also have varied, given the peculiarities of the Antiguan and Barbudan situation, our findings are consistent with those of many studies conducted elsewhere that have suggested that the most significant costs are incurred for hormonal therapy, surgical treatment, radiotherapy (external beam radiation and brachytherapy), and care for attendant post-treatment side effects [35,40]. As in those studies, in our study, the cost of these therapeutic items accounted for a considerable percentage of the annual direct medical cost burden of prostate cancer in Antigua and Barbuda.

Since this study estimated the economic burden of prostate cancer from the healthcare provider’s perspective and therefore, did not investigate any gains from a specific intervention, its findings could best serve as a reference point for further economic evaluations, including cost-effectiveness and cost–utility studies [18,26,41], e.g., an evaluation of the cost-effectiveness of robotic prostatectomy compared to radical prostatectomy as a functional intervention in patient outcomes [42]. Additionally, because the cost of managing early-stage prostate cancer could be prohibitive and the accompanying post-treatment side effects problematic [43], the findings highlight evidence that suggests that the major cost drivers in this study are sufficiently important to the healthcare provider to be considered in future cost containment strategies, a consideration that was noted in a similar study that looked at the cost burden of prostate cancer in Eswatini [7]. Future cost models could consider the costs of prostate cancer from the societal perspective, among a list of other useful methodologies for assessing and evaluating the economic burden of this disease, as well as for promoting meaningful resource usage at the levels of both policy and clinical practice in Antigua and Barbuda [7,26,44].

One strength of this study is that it highlighted the lack of a national register of costs, a national cancer registry, a national cancer plan, a nationally organized prostate-cancer-screening program, and legislative support mandating the reporting of cancer cases at various levels of patient care in Antigua and Barbuda [2,22]. A fully functioning cancer registry not only makes data readily available and ensures that data are organized but also facilitates assessment of the epidemiological burden of prostate cancer, including identifying prevalent cases and determining the local distribution of prostate cancer cases [22]. A further strength is that the study was restricted to the three main study sites mentioned in Section 2 and relied on (i) a robust epidemiological data collation process that included the use of a pretested data abstraction form that captured the sociodemographic, clinicopathological, and pharmacy data of prostate cancer cases; (ii) consultation of the literature to ensure a better understanding of the data collated on the parameters and components of cancer treatment; and (iii) participation in general discussions with health professionals, including the medical oncologist, the urologist, the pathologist, and the oncology nurses, with experience in the treatment and management of patients with prostate cancer at Sir Lester Bird Medical Center, Antigua and Barbuda. Thus, in addition to using a robust conventional health economics methodology to assess the cost burden, our consultations provided both scientific and contextual information that further shaped the costs of prostate cancer, thereby validating the study results [44]. This resulted in the accurate categorization of the cost components in this study, particularly given both the differences in the development of prostate cancer within our study population (including the tendency to develop biologically aggressive disease) [45] and the available cancer management options, which ranged from active surveillance to hormonal and systemic therapy [43,46]. Therefore, there is a high degree of confidence that the cost estimates in this study are a reliable representation of the current economic burden or cost of prostate cancer in Antigua and Barbuda.

This study has several limitations. A sizable proportion of the study’s population was between middle aged and old (55–74 years), which could have implications for our understanding of the estimates in populations of persons older than 74 years. However, this observation is attenuated because, in addition to the clinical stage, disease risk stratification, and the patient’s functional status, chronological age is a key factor in treatment decisions [41]. Our treatment costs, therefore, are reflective of an interplay of these scenarios. Another limitation is that our study focused only on the direct medical costs of prostate cancer and did not consider other cost elements, such as comorbidities, end-of-life care, and loss of productivity as a result of illness [26]. This would have resulted in our analysis presenting costs from one perspective. Thus, although our costs appear to be lower than what we would expect if we were to consider, for example, the costs of prostate cancer from a societal perspective [18], we believe they are still quite high and require attention. This is particularly true because our results are consistent with observations from other studies that have indicated that direct medical costs usually account for a significant portion of the costs of prostate cancer management [7,47]. Incorporating cost elements such as end-of-life care, the patient’s quality of life, and loss of productivity in future analyses of the cost of prostate cancer in Antigua and Barbuda could provide a more holistic view of the economic burden associated with this disease [41,48].

## 5. Conclusions

In conclusion, prostate cancer is a serious public health matter in Antigua and Barbuda. The evidence presented in this study suggests that the costs of managing this condition appear reasonable, given the local context. The most significant drivers of cost are surgery and hormonal therapy, used to treat stages II and III of the disease. Conducting further economic evaluations using either cost-effectiveness or cost–utility studies; undertaking future analysis of the costs of prostate cancer from the individual, employer, and societal perspectives; and considering the impact of indirect costs, especially those of comorbidities and lost productivity, on the costs of prostate cancer are recommended as important approaches to gaining a more comprehensive understanding of the economic burden of this cancer in Antigua and Barbuda. Collecting data from an established national cancer registry and a national register of costs would significantly benefit these approaches. In addition to these recommendations, however, our cost estimates are a reliable representation of the current economic burden and could, therefore, be considered a useful reference for policy makers in the Ministry of Health, Antigua and Barbuda, (i) for planning and making certain budget-related decisions regarding the treatment of prostate cancer in the country and (ii) for advocating and agreeing on appropriate cost containment measures. These measures could include reviewing the costs of prostatectomy options and revising procurement and selection practices for prostate-cancer-related drugs and other products.

## Figures and Tables

**Figure 1 ijerph-21-01527-f001:**
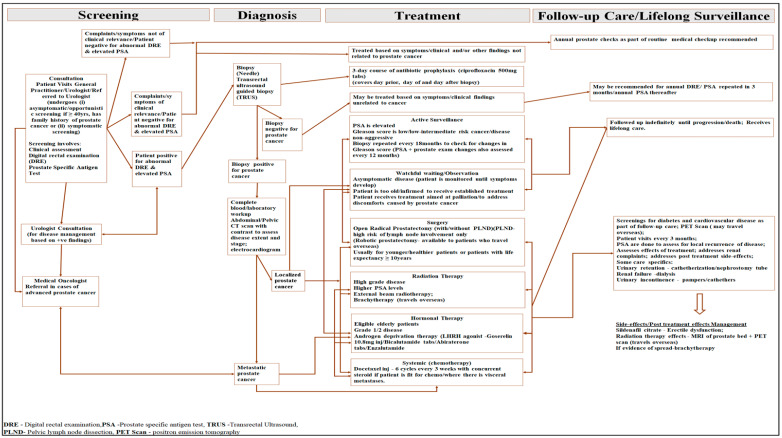
Schematic diagram showing the main components of care related to the management of prostate cancer in Antigua and Barbuda.

**Figure 2 ijerph-21-01527-f002:**
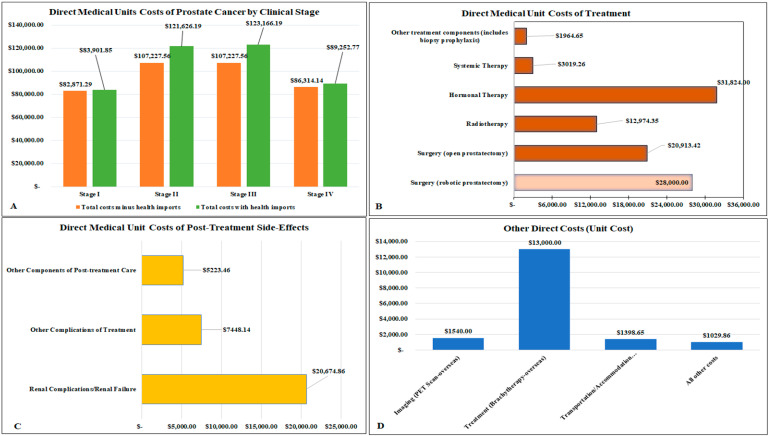
Direct medical unit costs by main costs drivers and per clinical stages (2021). (**A**) Clinical stage; (**B**) treatment; (**C**) post-treatment side-effects relieving procedures; (**D**) other direct costs.

**Figure 3 ijerph-21-01527-f003:**
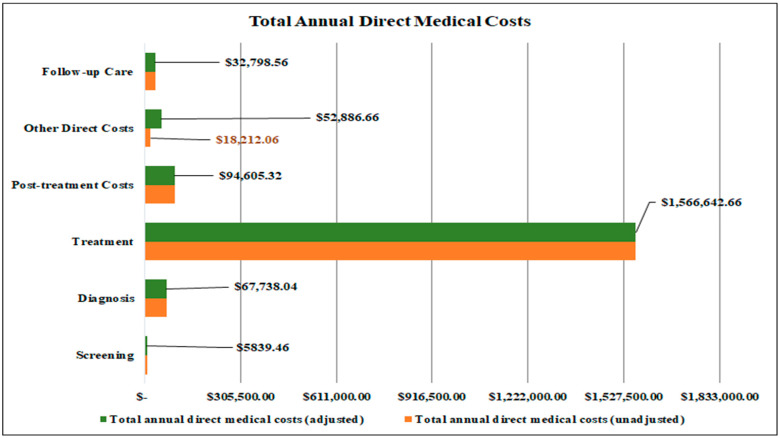
Total annual direct medical costs broken down by key cost parameters.

**Table 1 ijerph-21-01527-t001:** Data variables and source of costs regarding screening, diagnosis, treatment, and management of prostate cancer.

Data/Parameter	Data Source/Explanation	Price Source
**Estimated number of prostate cancer cases**	Combined number of cases abstracted from patient files of the Cancer Center of the Eastern Caribbean, Sir Lester Bird Medical Center, and the Medical Benefits Scheme for the period 2017–2021; Health Information Division, Ministry of Health.	N/A
**Screening**		
Consultation	Interview with experts from the Medical Benefits Scheme (review of reimbursement receipts presented for refunds and board-approved financing of care) and Sir Lester Bird Medical Center (billing and finance departments).	Market price
Clinical assessment		Market price
Digital rectal examination (DRE)		Market price
Prostate-specific antigen (PSA)	Medical Benefits Scheme (review of reimbursement receipts presented for refunds and board-approved financing of care), Antigua and Barbuda private laboratories charges.	Market price/private laboratory
**Diagnosis**		
Biopsy (TRUS)	Interview with experts, Medical Benefits Scheme (review of reimbursement receipts presented for refunds and board-approved financing of care), Sir Lester Bird Medical Center (billing and finance departments), Sir Lester Bird Medical Center Laboratory, Antigua and Barbuda private laboratories/clinic charges. Biopsy (transrectal ultrasound—TRUS); pathology, computed tomography (CT scan) of the abdominal/pelvis; chest X-ray; echocardiogram, complete blood workup following biopsy/PSA test, biopsy prophylaxis (antibiotic prophylaxis for three-day period to include day of biopsy).	Market price
Complete blood workup		Market price/private laboratory
Histopathology		Market price
Imaging studies (computed tomography (CT scan); X-ray)		Market price/private clinic
Biopsy prophylaxis		Market price
**Treatment**		
	Interviews with the consulting urologist, senior oncology nurses at the Cancer Center of the Eastern Caribbean and Sir Lester Bird Medical Center, interviews with pharmacists, Medical Benefits Scheme (review of reimbursement receipts presented for refunds and board-approved financing of care), Sir Lester Bird Medical Center (billing and finance departments), general discussion with a surgeon in private surgery, interview with manager of operating room at Sir Lester Bird Medical Center. Involves: Active surveillance, watchful waiting, radical prostatectomy (open and robotic, as per patient selection/preference), mainly external beam radiation for 5-week period; brachytherapy (overseas) for selected patients, 5-week period; androgen deprivation therapy (includes goserelin 10.8 mg injection IM once every 3 months, bicalutamide for local disease; abiraterone may be added in metastatic disease, mainly docetaxel injection with a steroid every 3 weeks for six cycles.	
Active surveillance		Market price
Watchful waiting		Market price
Surgery		Market price
Hormonal therapy		Market price/private supplier
Systemic therapy (chemotherapy)		Market price
**Post-treatment side-effects relieving procedures**		
Urinary retention	Interviews with the consulting urologist, senior oncology nurses at the Cancer Center of the Eastern Caribbean and Sir Lester Bird Medical Center, interviews with pharmacists, Medical Benefits Scheme (review of reimbursement receipts presented for refunds and board-approved financing of care), Sir Lester Bird Medical Center (billing and finance departments), general discussion with a surgeon in private surgery. Any procedure or approach required by a patient that received prostate-cancer-related treatment; may involve responses to urinary retention or incontinence, renal failure, erectile dysfunction, or other complications such as depression, fatigue, etc.	Market price
Urinary incontinence		Market price
Renal complications/renal failure		Market price
Erectile dysfunction		Market price
Lower urinary tract infections		Market price
Other complications of treatment		Market price
**Other direct costs**		
Nutrition counseling	Interviews with the consulting urologist, senior oncology nurses at the Cancer Center of the Eastern Caribbean and Sir Lester Bird Medical Center, interviews with pharmacists, interview with a nutritionist, Medical Benefits Scheme (review of reimbursement receipts presented for refunds and board-approved financing of care), Sir Lester Bird Medical Center (Billing and Finance departments). Most services are available to all patients locally; select patients have access to overseas positron emission tomography scanning (PET Scan) or brachytherapy services	Private: market price
Psychiatric/psychological counseling		Market price
Pharmacy Services		Private Pharmacy/market price
Imaging (PET scan—overseas)		MBS approved Financing/market price
Treatment (brachytherapy—overseas)		Private pharmaceutical Supplier/market price
Emergency kit		
Transportation/accommodation (imaging/treatment—overseas)		MBS approved Financing/market price
Transportation (local services related)		Market price
Overheads		Market price

**Table 2 ijerph-21-01527-t002:** Background characteristics of the population of cases of prostate cancer (2017–2021).

Variable Class	Characteristics	Prostate Cancer N = 109 N (%)
**Demographic**	**Age at Presentation**	
	Mean age (SD)	66.5 (7.8)
	Mean age 95% CI	65.0–68.0
	Median age	67.0 (11.0)
	Age range	46.0–90.0
	**Age Distribution**	
	45–54	3 (2.8)
	55–64	43 (39.5)
	65–74	46 (42.2)
	≥75	17 (15.6)
**Clinical**	**Clinical Stage**	
	I	9 (8.3)
	II	50 (45.9)
	III	34 (31.2)
	IV	16 (14.7)

**Table 3 ijerph-21-01527-t003:** Costs for screening, diagnosis, staging, treatment, and management of prostate cancer (stage I–IV) (estimated number of prostate cancer cases in one year, n = 19).

Screening, Diagnosis, Staging, Treatment, and Management Variables	Unit Costs (USD)	Unit Costs by Clinical Stage (USD)			
Estimated Number of Cases in a Single Year (N = 19)		I (n = 2)	II (n = 8)	III (n = 6)	IV (n = 3)
Screening					
Consultation (DRE, clinical assessment)	276.05	276.05	276.05	276.05	276.05
Laboratory test (PSA)	31.29	31.29	31.29	31.29	31.29
Diagnosis					
Biopsy (TRUS)	1339.76	1339.76	1339.76	1339.76	1339.76
Histopathology	478.49	478.49	478.49	478.49	478.49
Imaging studies (radiology)	1070.12	1070.12	1070.12	1070.12	1070.12
Complete blood workup	456.40	456.40	456.40	456.40	456.40
Biopsy prophylaxis	189.10	189.10	189.10	189.10	189.10
Treatment					
Active surveillance	1326.51	1326.51	0	0	0
Watchful waiting	449.04	449.04	0	0	0
Surgery (open radical prostatectomy)	20,913.42	0	20,913.42	20,913.42	0
Surgery (robotic prostatectomy—overseas) **	28,000.00	0	28,000.00	28,000.00	0
Radiotherapy	12,974.35	12,974.35	12,974.35	12,974.35	12,974.35
Hormonal therapy	31,824.00	31,824.00	31,824.00	31,824.00	31,824.00
Systemic therapy (chemotherapy)	3019.26	0	3019.26	3019.26	3019.26
Post-treatment side-effects relieving procedures					
Urinary retention	1447.24	1447.24	0	0	0
Urinary incontinence	2502.85	0	2502.85	2502.85	2502.85
Renal complications/renal failure	20,674.86	20,674.86	20,674.86	20,674.86	20,674.86
Erectile dysfunction	706.69	706.69	706.69	706.69	706.69
Lower urinary tract infections	125.00	125.00	125.00	125.00	125.00
Other complications of treatment	7448.14	7448.14	7448.14	7448.14	7448.14
Consultations	441.68	441.68	441.68	441.68	441.68
Other direct costs					
Nutrition counseling	100.00	100.00	100.00	100.00	100.00
Psychiatric/psychological counseling	128.82	128.82	128.82	128.82	128.82
Pharmacy services	89.99	89.99	89.99	89.99	89.99
Imaging (PET scan—overseas) **	1540.00	0	0	1540.00	1540.00
Treatment (brachytherapy—overseas) **	13,000.00	0	13,000.00	13,000.00	0
Emergency kit	112.94	0	112.94	112.94	112.94
Transportation/accommodation (imaging/treatment—overseas) **	1398.65	0	1398.65	1398.65	1398.65
Transportation (local services-related)	561.30	561.30	561.30	561.30	561.30
Overheads	36.81	36.81	36.81	36.81	36.81
Follow-up care					
Post-treatment consultations	1030.59	0	1030.59	1030.59	1030.59
Prostate-specific antigen test (PSA test)	125.14	125.14	125.14	125.14	125.14
Complete blood count with differential	268.69	268.69	268.69	268.69	268.69
Chemistry panel 7	33.13	33.13	33.13	33.13	33.13
Renal panel 1	143.55	143.55	143.55	143.55	143.55
Liver function tests	125.14	125.14	125.14	125.14	125.14
Total (unadjusted)	110,450.35	82,871.29	107,227.56	107,227.56	86,314.14
Total (if surgical option is robotic prostatectomy, with healthcare imports included)	133,475.56	83,901.85	128,712.77	130,252.77	89,252.77
Total (adjusted)	126,388.98	83,901.85	121,626.19	123,166.19	89,252.77

DRE: digital rectal exam; PSA: prostate-specific antigen; TRUS: transrectal ultrasound. ** Healthcare imports (cancer-related services and/or associated costs linked to overseas care). Unadjusted: the costs of healthcare imports are excluded from the analysis. Adjusted: the costs of healthcare imports, except robotic surgery, are included in the analysis.

**Table 4 ijerph-21-01527-t004:** Total annual costs estimation for prostate cancer (direct medical costs).

Parameter	Care Component/Procedures	Estimated Cases in One Year (N = 19)	Estimated Direct Medical Unit Cost 2021 (USD)	Total Annual Direct Medical Costs (USD)	Percentage of Total Costs (Adjusted)	Range (USD) ± 25%	
						Lower	Upper
Screening	Screening						
	Consultation	19	276.05	5244.95		3933.71	6556.19
	Laboratory test (PSA)	19	31.29	594.51		445.88	743.14
Subtotal				5839.46	0.32%	4379.60	7299.33
Diagnosis	Diagnosis						
	Biopsy	19	1339.76	25,455.44		19,091.58	31,819.30
	Complete blood workup	19	487.69	9266.11		6949.58	11,582.64
	Histopathology	19	478.49	9091.31		6818.48	11,364.14
	Imaging studies	19	1070.12	20,332.28		15,249.21	25,415.35
	Biopsy prophylaxis	19	189.10	3592.90		2694.68	4491.13
Subtotal				67,738.04	3.72%	50,803.53	84,672.55
Treatment	Treatment 1						
	Stage I	2	46,554.76	93,109.52		69,832.14	116,386.90
	Stage II	8	93,711.77	749,694.16		562,270.62	937,117.70
	Stage III	6	96,731.03	580,386.18		435,289.64	725,482.73
	Stage IV	3	47,817.60	143,452.80		107,589.60	179,316.00
Subtotal				1,566,642.66	86.06%	1,174,982.00	1,958,303.33
Post-treatment	Post-treatment side-effects relieving procedures						
	Urinary retention	3	1447.24	4341.72		3256.29	5427.15
	Urinary incontinence	3	2502.85	7508.55		5631.41	9385.69
	Renal complications/renal failure	1	20,674.86	20,674.86		15,506.15	25,843.58
	Erectile dysfunction	3	706.69	2120.07		1590.05	2650.09
	Lower urinary tract infections	3	125.00	375.00		281.25	468.75
	Other complications of treatment	8	7448.14	59,585.12		44,688.84	74,481.40
Subtotal				94,605.32	5.20%	70,953.99	118,256.65
Other Direct Costs	Other direct costs						
	Nutrition counseling	19	100.00	1900.00		1425.00	2375.00
	Psychiatric/psychological counseling	19	128.82	2447.58		1835.69	3059.48
	Pharmacy services	19	89.99	1709.81		1282.36	2137.26
	Imaging (PET scan—overseas) *	2	1540.00	3080.00		2310.00	3850.00
	Treatment (brachytherapy—overseas) *	2	13,000.00	26,000.00		19,500.00	32,500.00
	Emergency kit	7	112.94	790.58		592.94	988.23
	Transportation/accommodation (imaging/treatment—overseas) *	4	1398.65	5594.60		4195.95	6993.25
	Transportation (local services related)	19	561.30	10,664.70		7998.53	13,330.88
	Overheads	19	36.81	699.39		524.54	874.24
Subtotal				52,886.66	2.91%	39,665.00	66,108.33
Follow-up Care	Follow-up care						
	Post-treatment consultations	19	1030.59	19,581.21		14,685.91	24,476.51
	Prostate-specific antigen test (PSA test)	19	125.14	2377.66		1783.25	2972.08
	Complete blood count with differential	19	268.69	5105.11		3828.83	6381.39
	Chemistry panel 7	19	33.13	629.47		472.10	786.84
	Renal panel 1	19	143.55	2727.45		2045.59	3409.31
	Liver function tests	19	125.14	2377.66		1783.25	2972.08
Subtotal				32,798.56	1.80%	24,598.92	40,998.20
Total direct medical costs (unadjusted)				1,785,836.10		1,339,377.08	2,232,295.13
Total direct medical costs (adjusted)				1,820,510.70		1,365,383.03	2,275,638.38

* Healthcare imports (cancer-related services plus associated costs linked to overseas care). Unadjusted: the costs of healthcare imports are excluded from the analysis. Adjusted: the costs of healthcare imports, except overseas robotic surgery, are included in the analysis.

## Data Availability

All data generated or analyzed during this study are included in the article. The data are fully available without restrictions, and inquiries can be directed to the corresponding author.

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
