# Peer review of "The Economic Burden of Prostate Cancer in Antigua and Barbuda: A Prevalence-Based Cost-of-Illness Analysis from the Healthcare Provider Perspective"

_ijerph, 2024, doi:10.3390/ijerph21111527_

Round 1

Reviewer 1 Report

Comments and Suggestions for Authors

The Manuscript is a prevalence-based cost-of-illness analysis of prostate cancer oil Antigua and Barbuda. It is well-written but some points should be highlighted:

1) In the Methods section, is the geographic part useful to understand the way the study was conducted? Could this part be more useful if it was in the Discussion section?

2) In Methods section, the Authors should not show results, such as the number of patients considered (lines 104 - 125), or discuss the way the study was conducted (lines 181 - 187). Please, these parts should be moved in the right section;

3) Inclusion and exclusion criteria should be added;

4) Why did the Authors subtracted the 13 patients who died from the total? How did this choice influence the analysis?

Author Response

1. Summary

Thank you very much for taking the time to review this manuscript. We do express our appreciations to you for your comments and suggestions offered. It is our hope that the revised manuscript has addressed your concerns. We do look forward to hearing from you on this.

Please find the detailed responses below and the corresponding revisions/corrections highlighted in track changes in the re-submitted files.

2. Point-by-point response to Comments and Suggestions for Authors

Comments 1: In the Methods section, is the geographic part useful to understand the way the study was conducted? Could this part be more useful if it was in the Discussion section?

Response 1: Thank you for pointing this out.

Initially the authors thought that including the ‘geographic part’ about Antigua and Barbuda, would lend to readers appreciating not just the location and size of the country but also its layout in the context of the study.

Notwithstanding this thought, the authors have given much thought to the reviewers’ comment and have altered the ‘geographic part’. Much of it was removed.

In effect, the authors have revised subsection 2.1 of the methodology. This subsection was considerably reduced in keeping with a suggestion given by Reviewer #2.

“2.1.             Study area and population 

Antigua and Barbuda is the largest of the English-speaking Leeward Islands [9]. It is a twin-island state, with Antigua, the larger one, located 650 kilometers southeast of Puerto Rico, and Barbuda, the smaller one, located 48 kilometers north of Antigua [10]. The country’s population at its last census, held in 2011, was estimated at 85,567, with 40,986 males and 44,581 females and 80% (n=68,702) of the population being between the ages of 15 and 74 years [11].

As a member of the Eastern Caribbean Currency Union (ECCU), Antigua and Barbuda uses a common currency pegged to the United States dollar, the Eastern Caribbean dollar (XCD) [12,13].

With its monetary policy generally guided by the Eastern Caribbean Central Bank (ECCB), fiscally, one of the main contributors to the country’s gross domestic product (GDP) is tourism [14]. The country’s GDP per capita (current USD) was estimated at USD 17,178.50 for 2021, and the health expenditure per capita in the same year was approximately USD 923.41 [15].

Public healthcare is chiefly financed by the government-funded Medical Benefits Scheme (MBS) and allocations from the Ministry of Finance. Cancers are primarily diagnosed and managed at the country’s lone tertiary care hospital [16,17].

The study population consisted of men aged 18 years and older who had been diagnosed with prostate cancer between January 1, 2017, and December 31, 2021. The cancer cases were classified according to the International Classification of Diseases, 10th version (ICD-10) codes (C61) [18].”

Comments 2: In Methods section, the Authors should not show results, such as the number of patients considered (lines 104 - 125), or discuss the way the study was conducted (lines 181 - 187). Please, these parts should be moved in the right section;

Response 2: The authors have given much thought to the reviewers’ comments and wish to point out that since the study is a prevalence-based cost-of-illness study, then by demonstrating how prevalence cases are accounted for in the methodology is important in helping to define, for reasons of cost analysis, the number of cases requiring healthcare.

In this regard, the authors have inserted under subsection 2.4 (formerly 2.3), lines 131-134, the below information so as to lend clarity or definition to the number of cases considered for the analysis.

“For our cost analysis, we used an estimate of the number of prostate cancer cases in a single year in Antigua and Barbuda. This was calculated by subtracting the number of deaths from our total number of cases and dividing the answer by 5 [27].”

The authors posit that a similar approach was used by Ginindza and colleagues (Section Materials & Methods, subsection Cervical cancer, paragraph 3; Ref #27 on our list:

Ginindza TG, Sartorius B, Dlamini X, Östensson E. Cost analysis of Human Papillomavirus-related cervical diseases and genital warts in Swaziland. PLoS One 2017;12:e0177762. https://doi.org/10.1371/journal.pone.0177762.

Moreover, we have also included in the ‘Results’ section the following

“3.2      Estimate of Prostate Cancer Cases in a Single Year

The information obtained indicated that 13 of the 109 diagnosed cancer cases had died during the study period. We therefore estimated that there were 19 cases of prostate cancer on average in a single year in Antigua and Barbuda. This was calculated by subtracting the 13 deaths from the 109 cases diagnosed in the period and dividing the result (96) by 5.”

Regarding lines 181-187, the authors have since removed this from the subsection (now  2.4; previously 2.3) and retained information on the economic burden of prostate cancer in Sweden and Eswatini for incorporation in the discussion section as per suggestion/guidance of the Reviewer.

Comments 3: Inclusion and exclusion criteria should be added;

Response 3: The authors have considered the reviewers’ suggestion and have included this section as part of the Materials and Methods section (see below)

“2.2          Inclusion and exclusion criteria

In the absence of a cancer registry, information about prostate cancer cases diagnosed between January 1, 2017, and December 31, 2021, was obtained from medical records at the Sir Lester Bird Medical Centre, Antigua and Barbuda’s lone tertiary hospital; the Cancer Centre Eastern Caribbean (TCCEC), a public/private cancer facility; and the Medical Benefits Scheme (MBS), a statutory health organization. Information about prostate-cancer-related deaths was obtained from the Health Information Division, Ministry of Health, Antigua and Barbuda.

Cases of recurrent disease were excluded from the study.”

Comments 4: Why did the Authors subtracted the 13 patients who died from the total? How did this choice influence the analysis?

Response 4: Of the 109 prostate cancer cases diagnosed in the study period, information from the Health Information Division, Ministry of Health, Antigua and Barbuda indicated that they were 13 prostate cancer related deaths. Given the absence of a National Cancer Registry and hence a scarcity of prevalence data, the authors subtracted the 13 patients who died from the total number of cases to obtain a reasonable estimate of what the average prevalent cases would likely be for a single year in the study period.

The authors posit that a similar approach was considered by Ginindza and colleagues in paper reference DOI: 10.1371/journal.pone.0177762 under section ‘Materials and Methods’, paragraph 3 of subsection ‘Cervical cancer’.

In our study, use of this approach meant that the analysis yielded a most reliable estimate of average prevalent cases of 19 instead of 22 had we not accounted for the deaths.  

Authors Note

Kindly note that in addition to the edits done in respect of the comments and/or suggestions of the Reviewer, the authors have had the article subjected to English language editing so as to reduce grammatical errors. A copy of the certificate is included along with our submission of the revised manuscript.

Thank you

Reviewer 2 Report

Comments and Suggestions for Authors

The paper is well-organized, with a clear outline of the objectives, methodology, results, and discussion. The logical flow facilitates understanding, starting from the background of prostate cancer in the region to detailed cost analysis.

The Introduction section contains too many references, many of which are unnecessary. It should be shortened, the number of references reduced, and the content presented in clearer, more article-focused sentences. Much of the information currently in the Introduction would be better placed in the Discussion section. I recommend rewriting the Introduction section.

The '2.1. Study Area and Population' subsection is filled with unnecessary details and should be removed entirely. Only the key sentences can be included in the Discussion section.

The tables and figures are informative and well-presented, requiring no additional corrections.

All limitations should be compiled in a final paragraph at the end of the Discussion section.

What was your endpoint?

What are your recommendations and conclusions?

I suggest you to cite also: Vera-Álamo L, León-Medina P, Hernández-Flores CN, Armas-Molina JV, Artiles-Hernández JL. Comparison of quality of life after radical treatment in prostate cancer: radical prostatectomy versus external radiotherapy. Cir Cir. 2024;92(2):255-263. English. doi: 10.24875/CIRU.22000249. PMID: 38782388.             AND            Sánchez-Núñez JE, González-Cuenca E, Fernández-Noyola G, González-Bonilla EA, Doria-Lozano M, Rosas-Nava JE, Corona-Montes VE. Oncological and functional results after robot-assisted radical prostatectomy in high-risk prostate cancer patients. Cir Cir. 2022;90(S1):1-7. English. doi: 10.24875/CIRU.20001371. PMID: 35944100.

Comments on the Quality of English Language

Need some grammatical revisions

Author Response

1. Summary

Thank you very much for taking the time to review this manuscript. We do express our appreciations to you for your comments and suggestions offered. It is our hope that the revised manuscript has addressed your concerns. We do look forward to hearing from you on this.

Please find the detailed responses below and the corresponding revisions/corrections highlighted in track changes in the re-submitted files.

2. Point-by-point response to Comments and Suggestions for Authors

Comments 1: The Introduction section contains too many references, many of which are unnecessary. It should be shortened, the number of references reduced, and the content presented in clearer, more article-focused sentences. Much of the information currently in the Introduction would be better placed in the Discussion section. I recommend rewriting the Introduction section.

Response 1: Thank you for pointing this out. The authors have taken note of the reviewers’ comment and have rewritten the Introduction section. This resulted in the section being considerably shorter with a reduction in the number of references therein. The word count was reduced by some 220 words and reference count by 11.

Some of the information was considered for the discussion section.

Comments 2: The '2.1. Study Area and Population' subsection is filled with unnecessary details and should be removed entirely. Only the key sentences can be included in the Discussion section.

Response 2: The authors have since revised subsection 2.1 of the methodology. This subsection has been considerably reduced.

2.1.               Study area and population 

Antigua and Barbuda is the largest of the English-speaking Leeward Islands [9]. It is a twin-island state, with Antigua, the larger one, located 650 kilometers southeast of Puerto Rico, and Barbuda, the smaller one, located 48 kilometers north of Antigua [10]. The country’s population at its last census, held in 2011, was estimated at 85,567, with 40,986 males and 44,581 females and 80% (n=68,702) of the population being between the ages of 15 and 74 years [11].

As a member of the Eastern Caribbean Currency Union (ECCU), Antigua and Barbuda uses a common currency pegged to the United States dollar, the Eastern Caribbean dollar (XCD) [12,13].

With its monetary policy generally guided by the Eastern Caribbean Central Bank (ECCB), fiscally, one of the main contributors to the country’s gross domestic product (GDP) is tourism [14]. The country’s GDP per capita (current USD) was estimated at USD 17,178.50 for 2021, and the health expenditure per capita in the same year was approximately USD 923.41 [15].

Public healthcare is chiefly financed by the government-funded Medical Benefits Scheme (MBS) and allocations from the Ministry of Finance. Cancers are primarily diagnosed and managed at the country’s lone tertiary care hospital [16,17].

The study population consisted of men aged 18 years and older who had been diagnosed with prostate cancer between January 1, 2017, and December 31, 2021. The cancer cases were classified according to the International Classification of Diseases, 10th version (ICD-10) codes (C61) [18].”

Comments 3: The tables and figures are informative and well-presented, requiring no additional corrections.

Response 3: As per reviewers’ comment, no changes/corrections were made to the tables and figures.

Comments 4: All limitations should be compiled in a final paragraph at the end of the Discussion section.

Response 4: The authors have since restructured the discussion section in accordance with the reviewers’ suggestion.

Comments 5: What was your endpoint?

Response 5:  The authors have chosen to respond to this question from two points, namely that

(a)    This information is given in the first paragraph of the discussion section. Here the endpoint of the study, which in effect is the study purpose, was the total annual direct medical cost which provided an estimate of the economic burden or cost-of-illness of prostate cancer in Antigua and Barbuda in 2021. This was determined to be US$1.8 million.

The issue of how endpoint should be reported is given in page 659, subsection titled ‘Were Necessary Timeframes Specified?’ in the paper Larg and Moss 2011.

Allison Larg and John R. Moss. Cost-of-illness studies: A guide to critical evaluation. Pharmacoeconomics 2011;29:653–71. https://doi.org/10.2165/11588380-000000000-00000

https://doi.org/10.2165/11588380-000000000-00000

Since our study is a prevalence-based cost-of-illness study, then as Larg and Moss 2011 suggested, and I quote “Prevalence-based studies should specify the time period over which costs will be measured, usually a year.”

We feel that our estimate, the total annual direct medical cost of prostate cancer in Antigua and Barbuda, is a reliable representation of this cost.

And

(b)    Endpoint is interpreted to mean, now that the study has been done, what is next and/or how is this study useful for policy? What is the general contribution/significance of this work?

The authors posit that this work will be valuable as

(i)                  A reference point for the conduct of further economic evaluations regarding prostate cancer in Antigua and Barbuda

(ii)                A tool to be used by health policy makers in the Ministry of Health, Antigua and Barbuda for planning and making certain budget related decisions in respect of prostate cancer care in the country

(iii)              A tool to be used in advocating for a national register of costs or for getting health authorities to agree on instituting cost containment policies

Points of a similar nature were alluded to in papers 

Molinier L, Bauvin E, Combescure C, Castelli C, Rebillard X, Soulié M, et al. Methodological considerations in cost of prostate cancer studies: a systematic review. Value Health 2008;11:878–85. https://doi.org/10.1111/j.1524-4733.2008.00327.x.

and

Jo C. Cost-of-illness studies: concepts, scopes, and methods. Clin Mol Hepatol 2014;20:327–37. https://doi.org/10.3350/cmh.2014.20.4.327.

Comments 6: What are your recommendations and conclusions?        

Response 6: The conclusions section of the paper was rewritten to clearly identify a few recommendations emanating from this study and to strengthen the conclusion of itself.

Briefly given these recommendations are:

(i)                  Conducting further economic evaluations using either cost-effectiveness or cost-utility studies,

(ii)                Undertaking future cost-of-illness of prostate cancer from the individual, employer and societal perspectives,

(iii)              Considering the impact of indirect costs, especially that of comorbidities and lost productivity in future cost-of-illness of prostate cancer.

Said recommendations are highlighted in the revised Conclusions section (see below):

In conclusion, prostate cancer is a serious public health matter in Antigua and Barbuda. The evidence presented in this study suggests that the costs of managing this condition appear reasonable, given the local context. The most significant drivers of cost are surgery and hormonal therapy, used also to treat stages II and III of the disease. Conducting further economic evaluations using either cost-effectiveness or cost–utility studies; undertaking future analysis of the cost of illness of prostate cancer from the individual, employer, and societal perspectives; and considering the impact of indirect costs, especially those of comorbidities and lost productivity, on the cost of illness of prostate cancer are recommended as important approaches to gaining a more comprehensive understanding of the economic burden of this cancer in Antigua and Barbuda. Collecting data from an established national cancer registry and a national register of costs would significantly benefit these approaches. In addition to these recommendations, however, our cost estimates are a reliable representation of the current economic burden and could, therefore, be considered a useful reference for policy makers in the Ministry of Health, Antigua and Barbuda, (i) for planning and making certain budget-related decisions regarding the treatment of prostate cancer in the country and (ii) for advocating and agreeing on appropriate cost containment measures. These measures could include reviewing the costs of prostatectomy options and revising procurement and selection practices for prostate-cancer-related drugs and other products.

Thus, both recommendations and overarching conclusions are now accounted for in the Conclusions section of the article.

Comments 7: I suggest you to cite also:

Vera-Álamo L, León-Medina P, Hernández-Flores CN, Armas-Molina JV, Artiles-Hernández JL. Comparison of quality of life after radical treatment in prostate cancer: radical prostatectomy versus external radiotherapy. Cir Cir. 2024;92(2):255-263. English. doi: 10.24875/CIRU.22000249. PMID: 38782388.   

Response 7: This article has been cited as Ref # 41 in the Discussion section of the manuscript. 

Comments 8: I suggest you to cite also:

Sánchez-Núñez JE, González-Cuenca E, Fernández-Noyola G, González-Bonilla EA, Doria-Lozano M, Rosas-Nava JE, Corona-Montes VE. Oncological and functional results after robot-assisted radical prostatectomy in high-risk prostate cancer patients. Cir Cir. 2022;90(S1):1-7. English. doi: 10.24875/CIRU.20001371. PMID: 35944100.

Response 8: This article has been cited as Ref # 42 in the Discussion section of the manuscript. 

Comments 9:  Comments on the Quality of English Language- Need some grammatical revisions

Response 9: The revised manuscript was referred to MDPI editing services for English Language editing. See certificate in attachment.

Additional notes

Kindly note that the article was initially revised by the author as per comments and suggestions of the reviewers. After that it was sent to the MDPI editing services for English Language editing. Following receipt of the English edited version, further changes were made to the article.

Note that the tables and figures were not subjected to English-language editing.

The Abstract received English-language editing, however, the author had to re-edit same to ensure that the maximum word count of 200 words is maintained.

Thank you.

Round 2

Reviewer 1 Report

Comments and Suggestions for Authors

The Authors followed Reviewers' suggestions and the Manuscript is suitable to be published.

Reviewer 2 Report

Comments and Suggestions for Authors

Accept

Comments on the Quality of English Language

Good